# Preparation of a Series of Pd@UIO-66 by a Double-Solvent Method and Its Catalytic Performance for Toluene Oxidation

**DOI:** 10.3390/ma13010088

**Published:** 2019-12-23

**Authors:** Chuanying Wei, Haili Hou, Ermo Wang, Min Lu

**Affiliations:** School of Chemical Engineering, Northeast Electric Power University, Jilin 132000, China; 13844204685@163.com (C.W.); h18704329087@163.com (H.H.); asdfg8596@126.com (E.W.)

**Keywords:** Toluene oxidation, Pd@UIO-66, Double-solvent method

## Abstract

This paper reports on the preparation, characterization, and catalytic properties of the Pd@UIO-66 for toluene oxidation. The samples are prepared by the double-solvent method to form catalysts with large specific surface area, highly dispersed Pd^0^ (Elemental palladium) and abundant adsorbed oxygen, which are characterized by X-ray Photoelectron Spectroscopy (XPS), Brunauer-Emmett-Teller (BET) and Transmission Electron Microscopy (TEM). The results show that as the Pd content increases, the adsorbed oxygen content further increases, but at the same time Pd^0^ will agglomerate and lose some active sites, which will affect its catalytic performance. While 0.2%Pd@UIO-66 has the highest concentration of Pd^0^, the result shows it has the best catalytic activity and the T_90_ temperature is 210 °C.

## 1. Introduction

Volatile Organic Compounds (VOCs) are a series of substances that have a negative impact on the environment, mainly from vehicle emissions and industrial production [1,2,3,4,5]. Among many organic substances, toluene has attracted attention as one of the most widely affected volatile organic compounds [6]. Humans have already invented many methods to control them since recent decades. Among these methods, the combustion catalysis is the simplest and most feasible [7,8,9,10]. The combustion catalysis is the deep degradation of VOCs to CO_2_ and H_2_O at low temperatures (200–500 °C) by means of a catalyst [11] and catalysts play a vital role in combustion catalysis. At this stage, catalysts are mainly divided into supported noble metal catalysts (SNMCs) and supported transition metal catalysts (TMOs). Compared with TOMs, SNMCs have a better low temperature activity and easy regeneration. Noteworthy, the supported Pd materials are most widely investigated, which is probably due to the fact that it can make VOCs destroyed more easily at low temperatures [12,13,14,15].

For Pd-based catalysts, the dispersion of the active ingredient and the interaction with the support are especially critical for catalytic efficiency [16,17]; therefore, research on the support is very important. In addition, the supports in the Pd material are usually SiO_2_ andγ-Al_2_O_3_, but molecular sieves and metal oxides are used as well, including ZSM-5, SBA-15, TiO_2,_ and CeO_2_, to enhance the dispersion of active sites and aid reactant adsorption [17,18,19,20,21]. Therefore, to some extent, the supports with large specific surface area and strong adsorption will be more helpful to the performance of Pd.

Metal-organic frameworks (MOFs) are emerging materials with surprising specific surface area, adjustable pore structure and high concentration of metal ions, which are mainly used in gas adsorption and separation, sensors, and catalysis in many fields [22,23,24,25]. Among many series of materials, the UIO-66 is a metal-organic framework with Zr as the metal center and terephthalic acid as the organic ligand; it has a large specific surface area and a developed pore structure, is easy to chemically functionalize, and has excellent mechanical stability and heat resistance. Due to the above characteristics, UIO-66 and its derived UIO-66 series have received extensive attention in the field of catalysis. As a support for Pd, the UIO-66 is mainly used in Suzuki–Miyaura coupling reaction, hydrogenation catalysis and catalytic upgrading, etc. [23,26,27]. Luz et al. [28] apply the Pd@UIO-66-type to selective hydrogenation catalysis. The structure of UIO-66 combined with chemical vapor infiltration forms Pd-Nano particles (NPs), providing the efficiently selective activity. Jiang and co-workers [23] prepare UIO-66 sealed nano-palladium for efficient and continuous catalytic upgrade of ethanol to n-butanol. The close synergy between highly dispersed Pd and unsaturated Zr sites is the reason of high activity. Through the above reports, it can be found that the preparation method of Pd@UIO-66 needs to follow strict conditions, and Xu et al. [29] realize the simple preparation of Pd@MOF by the double-solvent method, which uses hydrophilic pores of MOFs. The double-solvent include the hydrophilic solvent (water) and the hydrophobic solvent (n-hexane). The former disperses the metal precursor, the volume is less than or equal to the pore volume, and it can be absorbed by the hydrophilic pores, while the latter provides the suspension adsorbent and promotes the impregnation. The result of this simple method is the reduction of deposits on the outer surface. Using the pore space of MOFs to better disperse Pd, this method is beneficial to the formation of noble metal NPs. Few studies have investigated the uses of Metal-organic frameworks (MOFs) impregnated with noble metals for the catalytic oxidation of volatile organic compounds. As a support, MOFs can improve the dispersion and enhance adsorption of NPs. Therefore, Pd@UIO-66 application may have the amazing performance in the VOCs catalysis.

In this research, Pd@UIO-66 was prepared as a catalyst by the double-solvent method, and its feasibility and catalytic efficiency as a new catalytic material in the catalytic reaction of toluene were studied. The stability and durability of the catalyst were also evaluated. Correspondingly, the relationship between catalyst structure and catalyst activity is characterized the material by N_2_ adsorption-desorption isotherm, and the data for analysis was obtained using the Brunauer-Emmett-Teller (BET) model, X-ray Diffraction (XRD), Transmission Electron Microscopy (TEM), X-ray Photoelectron Spectroscopy (XPS), and Fourier Transform Infrared Spectroscopy(FTIR).

## 2. Materials and Methods 

### 2.1. Preparation of Catalysts

Preparation of UIO-66: In this study, the UIO-66 was prepared from the hydrothermal method in accordance with the procedure reported by H. Wu et al [30]. The standard MOFs hydrothermal synthesis was performed by dissolving ZrCl_4_ (0.053 g, 0.227 mmol) and terephthalic acid (0.034 g, 0.227 mmol of H_2_BDC) in N,N-dimethylformamide (24.9 g, 340 mmol of DMF) at the room temperature with sufficient agitation to obtain a mixture. The mixture was transferred to a 100 mL polytetrafluoroethylene reactor and reacted at 120 °C for 24 h. After cooling to room temperature, the white precipitate was filtered, then washed repeatedly with DMF and methanol, finally dried at 100 °C to obtain a white product.

Preparation of Pd@UIO-66: Pd@UIO-66 synthesis adopts the double-solvent method, and the hydrophilicity of the MOFs cavity is used to disperse Pd on the inner surface as much as possible, thereby improving the use efficiency of the support and increasing the dispersion degree of the active ingredient. 0.5 g UIO-66 were dispersed in 100 mL n-hexane and sonicated for 1 h. In addition, 0.1875 mol/L aqueous solution of palladium nitrate was used as a precursor to prepare elemental palladium (Pd^0^). 12 µL Palladium nitrate solution was dropwise added to the dispersed UIO-66 in hexane and stirred for 2 h, then dried at the room temperature. The solid was dried in a vacuum drying oven at 100 °C, transferred to a tube furnace, and reduced at 200 °C for 5 h in a 10 vol % H_2_/N_2_ atmosphere to obtain 0.1 wt % Pd@UIO-66, hereinafter referred to as 0.1% Pd@U, 0.2% Pd@U, 1% Pd@U and 2% M@U materials were prepared according to the above method.

### 2.2. Catalyst Characterization

XRD patterns were recorded on a Rigaku Smart Lab 9kW diffractometer (Tokyo, Japan)with Cu-Kα radiation (λ = 1.5418 Å) in the 5–55° range with a step size of 0.02°. N_2_ adsorption-desorption isotherms were performed at 77 K on a micromeritics ASAP 2460 instrument (Micromeritics, Norcross, GA, USA) in a static mode; all samples were degassed at 623 K for 4 h. The specific surface area was calculated by the BET equation, and the pore volume and average pore diameter were calculated based on the Barrett-Joyner-Halenda (BJH) method. XPS measurements were performed on Thermo Fisher Scientific Esca Lab 250Xi instrument (Shanghai, China). The peak positions were corrected by using the containment carbon (C 1s peak = 284.8 eV). Changes in the organic framework of the material were characterized by FTIR and experiments were performed on a Nicolet iS50 FTIR spectrometer (Shanghai, China). An inductively coupled plasma optical emission spectrometer (ICPOES, Optima 2000DV, MA, USA) was employed to determine the overall Ce content in the prepared catalyst.

### 2.3. Catalytic Activity Measurement

The catalytic combustion of VOCs was carried out in a fixed-bed quartz reactor using a 0.10 g catalyst of 40–60 mesh. The reactant mixture contained 1000 ppm toluene and 20 vol % O_2_ balanced in N_2_ (balance gas) with a total flow of 100 mL min^−1^, giving a GHSV of 60,000 mL (g⋅h)^−1^. Before the start of each experiment, the catalyst needs to be pretreated for 1 h in the real reaction gas in order to overcome the overestimation caused by toluene adsorption. A gas chromatograph (Agilent 7890A) equipped with a flame ionization detector (FID, using a column of Porapack-Q/molecular sieve5A, 2 m in length) and a thermal conductivity detector (TCD, using a column of RT-QPlotdivinylbenzene PLOT, 30 m in length) were used to monitor the trend of the toluene concentration online (Appendix A). The toluene conversion (%) was calculated according to the following equation:$$Toluene\;conversion=\frac{{\left[Toluene\right]}_{in}-{\left[Toluene\right]}_{out}}{{\left[Toulene\right]}_{in}}\times 100\%$$

Arrhenius plots the activation energy from the toluene rate of the sample.

## 3. Results and Discussion

### 3.1. Material Structure and Physical—Chemical Properties

Figure 1 represents the Pd@UIO-66 catalyst prepared by UIO-66 as a support at N_2_ adsorption-desorption isotherm at 77 K. All solids exhibit a microporous structure. Obviously, at 1% Pd@U and 2% Pd@U, the N_2_ adsorption volume rises sharply and exhibits a relatively large hysteresis loop at very low relative pressure [31,32]. The N_2_ adsorption-desorption isotherm curve of UIO-66 is collected in Appendix A. The value of the pore volume in Appendix A can reflect this phenomenon more clearly. The pore volume and specific surface area of 0.1% Pd@U and 0.2% Pd@U are much larger than 1% Pd@U and 2% Pd@U. The former is close to the data of pure UIO-66. Although the specific surface area and pore volume differed greatly, there was no significant change in the pore size of the sample, which indicates that the double-solvent method is mild to the UIO-66 structure and does not affect its basic microporous structure. The larger specific surface area, the more favorable for toluene catalysis, so low loading of 0.1% Pd@U and 0.2% Pd@U may have better catalytic efficiency relative to high load samples [33].

The XRD patterns of UIO-66 and Pd@UIO-66 are shown in Figure 2. It can be seen that UIO-66 has obvious peaks, which is consistent with previous reports [34], which also shows that UIO-66 has the excellent crystallinity. As shown in Figure 2, the Pd@UIO-66 diffraction peak is the same as UIO-66, and the diffraction peak of Pd is not observed, which indicates that the Pd NPs content is low, and the dispersion is high, which is well wrapped by the pores of UIO-66 [29]. However, there is an interesting phenomenon that the diffraction peaks of the 1% Pd@U and 2% Pd@U samples are significantly broadened, which indicates that although the two samples have a further effect on the crystallinity of UIO-66 due to the further increase in Pd content. Obviously, the double-solvent method as a mild preparation method does not affect the crystal structure and the basic microporous structure of UIO-66, and greatly retains the structural characteristics of UIO-66 as a support.

The structure and the Pd distribution of the Pd@UIO-66 catalysts were analyzed by TEM microscopy. Figure 3a–d shows TEM images of 0.1% Pd@U to 2% Pd@U catalyst, respectively. The Pd particle image appears as black spots, appears to be scattered and spherical [18,23]. It can be found that as the palladium concentration increases, the size of NPs also increases. It is worth noting that as the metal palladium concentration increases, larger particles of Pd NPs appear in the 1% and 2% samples, and it can be observed from Figure 3c,d that the majority of the particles are located on the outer surface of the material. This may be detrimental to the catalytic performance of the material. For low concentration load samples, the large surface area of UIO-66 combined with the introduction of Pd by the double-solvent method not only forms highly dispersing of Pd NPs, but also disperses Pd in a large amount on the inner surface of UIO-66, which will improve the specific surface area use of metal particles [23]. Highly dispersed Pd NPs have a major positive impact on the catalytic performance of toluene.

To further understand the chemical states of Pd and O on the surface of Pd@UIO-66, XPS characterization was performed (Figure 4 and Appendix A). The binding energy values of Pd3d^3/2^ are shown in Figure 4A, and Pd^0^ (333.38–333.57 eV) and Pd^2+^ (335.28–335.85 eV) were observed [12,17]. At the same time, after comparing among the four samples, it can be found that as the Pd content increases, the binding energy migrates to high energy. For palladium species, the metal palladium site is very active in the decomposition of VOCs, so it plays an important role in adsorbing and decomposing toluene to form various secondary products such as benzaldehyde and benzoic acid. Correspondingly, PdO provides an additional surface oxygen source that is more susceptible to oxidation of secondary products. Finally, the toluene is completely oxidized to CO_2_ and H_2_O [35,36,37]. In this regard, the 0.2% Pd@U sample with the highest Pd^0^ ratio (95.13%) may be beneficial to the catalytic direction of toluene.

As shown in Figure 4B, in the 530.10–530.18 eV interval is lattice oxygen, 531.85–531.91 eV is adsorbed oxygen. In addition, a high concentration of adsorbed oxygen is exhibited, which is attributed to the fact that the oxygen-rich group in the UIO-66 material adsorbs to the unsaturated Zr-O site and forms adsorbed oxygen under certain conditions. In addition, as the concentration of palladium increases, the concentration of adsorbed oxygen is in an ascending order (2% Pd@U > 1% Pd@U > 0.2% Pd@U > 0.1% Pd@U) [23]. The binding energy image of Zr3d is collected in Appendix A. It can be observed from the figure that the binding energy of Zr shifts toward high energy, which indicates that as the palladium increases, the outer electrons of Zr decrease. Among them, 0.2% Pd@U sample has the highest binding energy of Pd, Zr and lattice oxygen relative to other materials. Generally, this is due to the strong electronic interaction between Pd-O-Zr which promotes electron transfer between the active ingredient and the support [38].

The oxidation mechanism of Pd species is mainly the L–H (Langmuir–Hinshelwood) model, i.e., the two reactants are simultaneously adsorbed on the same active site for reaction. Pd^0^ acts as an active site for the adsorption and dissociation of toluene, while a high concentration of adsorbed oxygen helps to continue to oxidize the decomposed material to form H_2_O and CO_2_ [14,39]. Therefore, the content of Pd^0^ is more critical for the first decomposition of toluene relative to the oxygen concentration. Moreover, the oxidation mechanism of the Pd catalyst generally follows the L-H model [40]. As such, in the presence of high concentrations of Pd^0^, high concentrations of adsorbed oxygen reactions are advantageous for the reaction.

### 3.2. Catalytic Performance

In the current study, the performance of the catalyst is tested by a toluene-catalyzed oxidation reaction in the range of 170–270 °C. The blank experiment catalyzed pure UIO-66 without metals using toluene. The activity is not satisfactory, and the catalytic performance is always lower than 10% in the reaction temperature range (Appendix A). The activity was proposed in Figure 5A. Toluene was completely oxidized to water and carbon dioxide by each sample, and no other by-products were found during the reaction, which explained that the system maintains a 99.5% carbon balance in each run. It can be seen intuitively that all four catalysts have typical S-shaped curve activity, and the toluene catalytic concentration is also enhanced with the increasing temperature. The T_50_ and T_90_ temperatures of the four catalysts are collected in Appendix A. For 0.1% Pd@U, 0.2% Pd@U, 1% Pd@U and 2% Pd@U, the T_50_ values are 232, 210, 236 and 238 °C and the T_90_ values are 238, 217, 246 and 249 °C, respectively. Therefore, the catalytic activity of toluene oxidation increased in the following order: 2% Pd@U < 1% Pd@U < 0.1% Pd@U < 0.2% Pd@U. This fact indicates that the catalytic efficiency of Pd-based SNMCs is affected by Pd^0^ and O. The high concentration and high dispersion of Pd^0^ provide abundant adsorption sites and accelerate the decomposition of toluene. This can explain that 1% Pd@U has a high concentration of Pd^0^ but the activity is less than 0.1% Pd@U. The apparent activation energy of each sample was calculated through the Arrhenius curve and is shown in Figure 5B. To prevent the sample from being affected by internal diffusion, the interval was selected below 20% to plot the Arrhenius curve and the apparent activation energies of each sample was calculated to be 76.88 kJ mol^−1^ (0.1% Pd@U), 69.84 kJ mol^−1^ (0.2% Pd@U),85.18 kJ mol^−1^ (1% Pd@U) and 87.56 kJ mol^−1^ (2% Pd@U). The data corresponds to the catalytic oxidation performance of the sample, 0.2% has the lowest activation energy and is the best activity, which also corresponds to the previous characterization.

### 3.3. Stability and Durability Test

It was observed by XRD image (Appendix A) that there was no change in the characteristic diffraction peak of UIO-66 before or after the reaction. In addition, the specific organic framework changes were exhibited by FTIR (Appendix A), the characteristic peak of the carbon-oxygen double bond of the carboxyl group at 1661 cm^−1^, the absorption peak of 1575 cm^−1^ and 1393 cm^−1^ is COO-, the characteristic peak of the benzene ring is 1500 cm^−1^, and the vibration peak of 749 cm^−1^ is derived from the vibration of Zr-O [41]. It was found that the carboxyl group was slightly red-shifted after the reaction, indicating that partial deligandation, which was attributed to the removal of some of the coordination functional groups by UIO-66 at high temperatures. However, the other relatively basic structure has not changed, which indicates that the basic structure of the material remains stable at the reaction temperature, which is consistent with the XRD data [42,43]. The stability and durability of the catalyst were evaluated under conditions of a toluene conversion of 18%. Appendix A shows catalyst conversion needs for over a 20-hour-run time (200 °C). During the experiment, there was no significant change in the conversion rate, which indicated that the material maintained relatively good stability and durability during the whole process [44].

## 4. Conclusions

Pd@UIO-66 was prepared by the double-solvent method to study the effect of different Pd loading concentrations on the catalytic performance of toluene. The results show that 0.2% Pd@UIO-66 has the best performance, which is attributed to its large specific surface, high dispersion of Pd, and abundant adsorption oxygen. Furthermore, there is a strong electronic interaction between the unsaturated Zr-O bond and Pd in UIO-66, which promotes the electron transfer of the carrier and the active component and increases the activity of Pd, while the high concentration of elemental palladium toluene adsorption and decomposition also play an important role. Together with these factors, 0.2%Pd@UIO-66 exhibits excellent low temperature catalytic ability.

## Figures and Tables

**Figure 1 materials-13-00088-f001:**
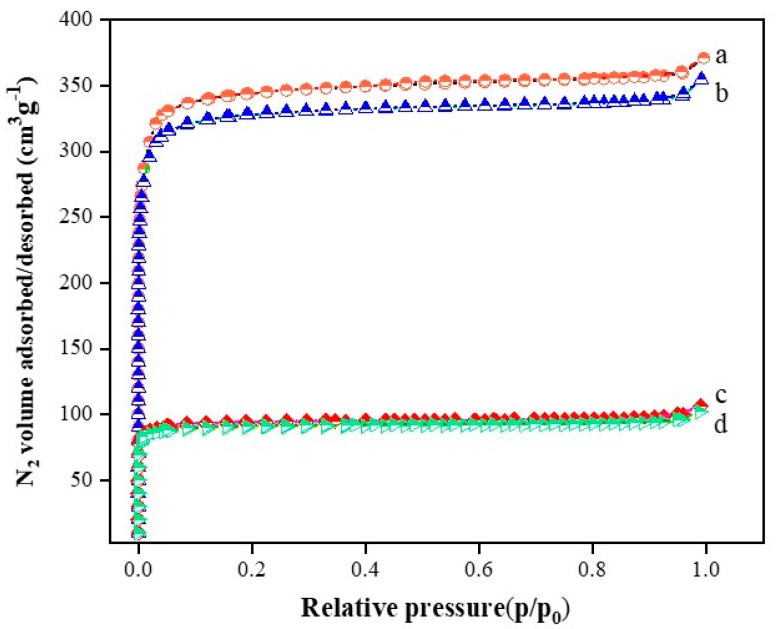
N_2_ adsorption/desorption analysis of (**a**) 0.1% Pd@U (**b**) 0.2% Pd@U (**c**) 1% Pd@U (**d**) 2% Pd@U.

**Figure 2 materials-13-00088-f002:**
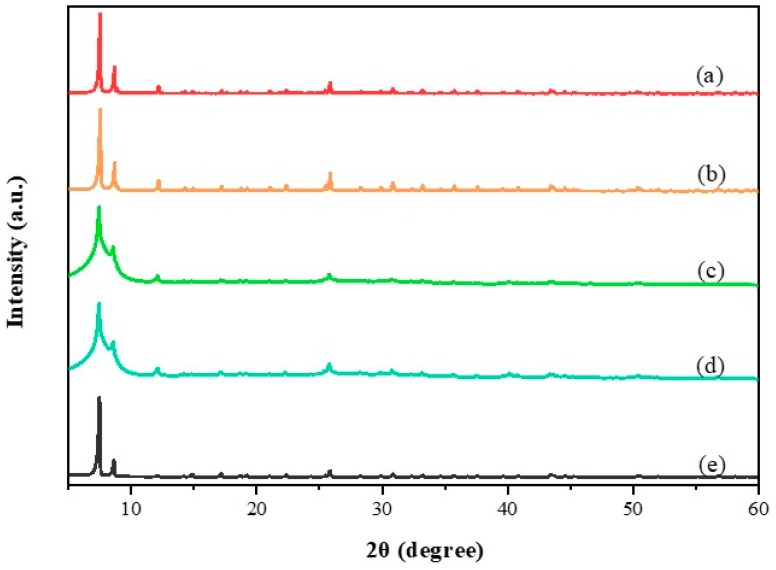
XRD patterns of (**a**) 0.1% Pd@U (**b**) 0.2% Pd@U (**c**) 1% Pd@U (**d**) 2% Pd@U (**e**) UIO-66.

**Figure 3 materials-13-00088-f003:**
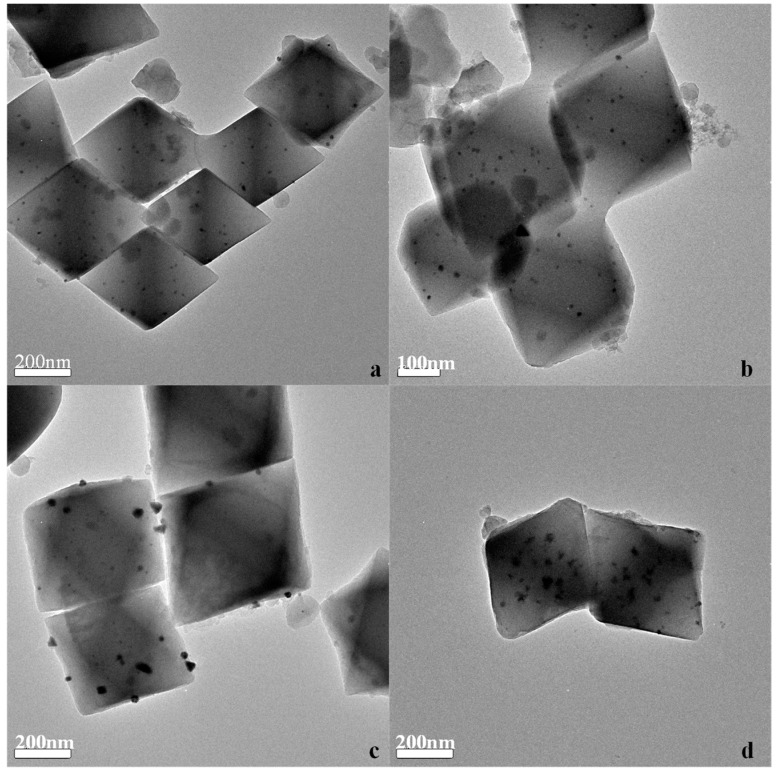
TEM images of (**a**) 0.1% Pd@U (**b**) 0.2% Pd@U (**c**) 1% Pd@U (**d**) 2% Pd@U.

**Figure 4 materials-13-00088-f004:**
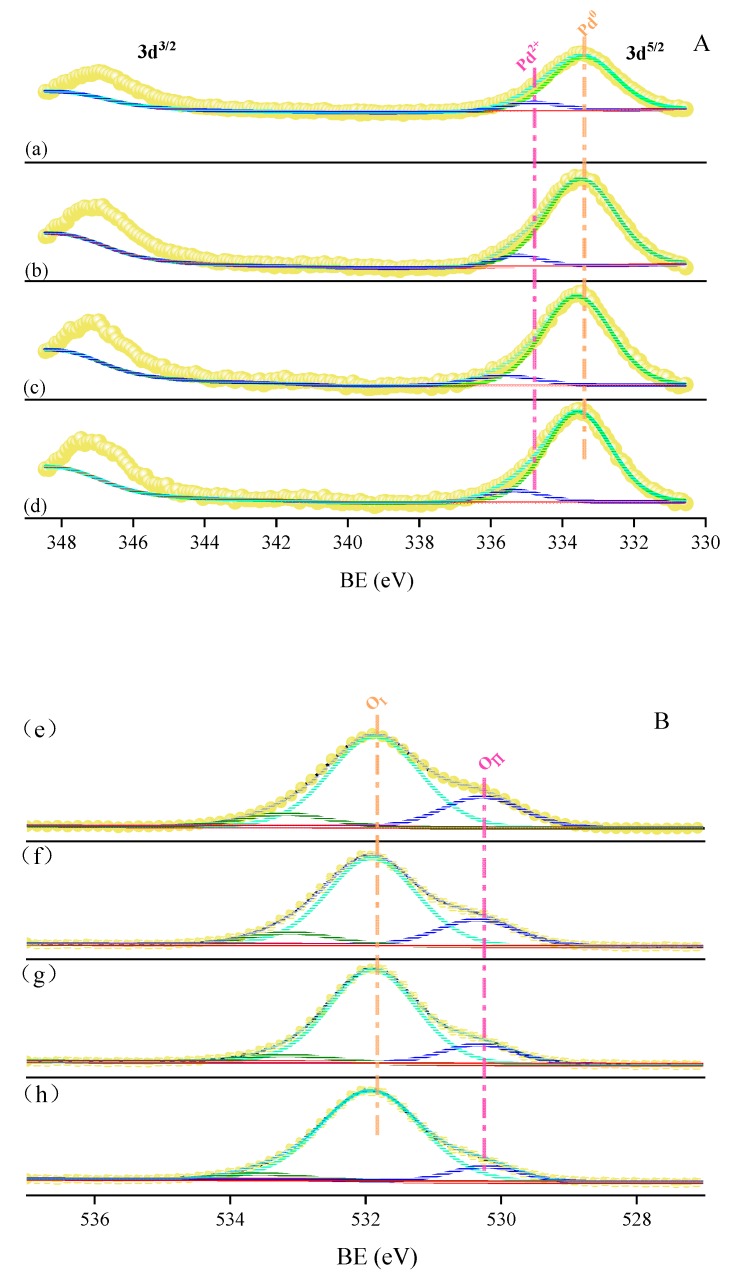
XPS spectra of (**A**) Pd3d (**a**) 0.1% Pd@U (**b**) 0.2% Pd@U (**c**) 1% Pd@U (**d**) 2% Pd@U.And (B) O1s (**e**) 0.1% Pd@U (**f**) 0.2% Pd@U (**g**) 1% Pd@U (**h**) 2% Pd@U.

**Figure 5 materials-13-00088-f005:**
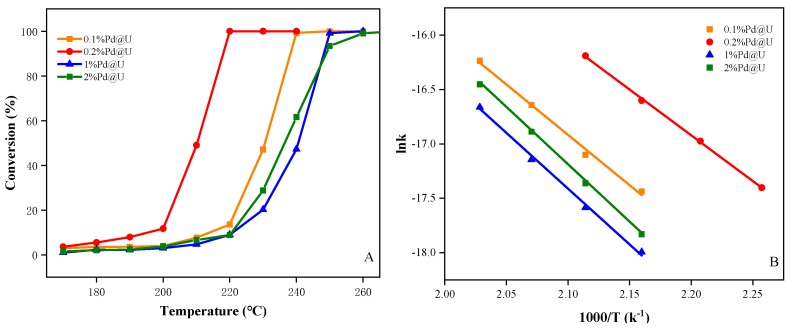
(**A**) Toluene conversion and (**B**) Arrhenius plots for toluene oxidation over the samples under the conditions of toluene concentration = 1000 ppm, 20 vol % O_2_, and WHSV = 60,000 mL (g⋅h)^−1^.

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
