# Peer review of "Preparation of a Series of Pd@UIO-66 by a Double-Solvent Method and Its Catalytic Performance for Toluene Oxidation"

_materials, 2019, doi:10.3390/ma13010088_

Round 1
Reviewer 1 Report
The paper discusses the preparation and use of the Pd @ UIO-66 catalyst, in thermal conditions.
In most cases these catalysts are used in photo-catalytic conditions therefore this work represents an interesting application, albeit a limited one.
The problems of this work include the lack of a study on the TON of the catalyst and its lifetime. In fact, I disagree that the simple analysis of durability and spectra after reaction can be exhaustive.
Furthermore, the background reaction without catalyst is missing, therefore it is not possible to evaluate the effects of combustion as it is.
Finally, I will suggest extending the method also to a saturated compound such as the hexane to verify the efficiency of catalyst
Reviewer 2 Report
Comments for materials-665014
This manuscript presents a new UIO-66-supported noble metal catalyst for the oxidation of toluene as an environmental threatening volatile hydrocarbon. The catalyst was characterized using BET, XRD, XPS, FTIR, and TEM analyses. Although the catalyst showed relatively good activity towards the oxidation reaction, the present version of the manuscript suffers from a number of shortcomings that must be addressed before reconsideration for publication. Here are my main concerns that I would like to see addressed in a revised version:
1- In the abstract, the authors must briefly explain in parenthesis what Pd0 stands for.
2- More information is needed in introduction about UIO-66, its structure, chemical composition, and methods of synthesis.
3- More broad information must be presented about other catalysts for the conversion of BTX hydrocarbons. I would recommend that the authors cite the following articles to improve the manuscript:
Fuel: 243 (2019) 469-477. Applied Catalysis A, General: 558 (2018) 109-121.
4- All abbreviations (such as DMF) must be expanded as they first appear in the manuscript.
5- It is not clear in the introduction section why the authors have decided to choose toluene oxidation as a means of catalyst performance evaluation.
6- I would recommend that a schematic diagram of the experimental setup (i.e. the reactor) be included in the manuscript or in supplemental information.
7- As part of catalyst preparation, the catalyst precursor has been reduced at 200 °C. However, neither TPR analysis has been reported to clarify the optimum reduction temperature, nor any proper reference(s) has been cited.
8- N2 adsorption/desorption profile of the catalyst support (UIO-66) must be presented in Fig. 1 for comparison.
9- Blank experiment using catalyst support for the oxidation of toluene is missing.
Reviewer 3 Report
The manuscript has been written well for the introduction, methods and results. The manuscript has good data and interpretations. However, the manuscript needs minor revision on the following:
English and grammar All abbreviations must be spelled out as and when it first appears in the text . Examples include: (a) UIO-66 and XPS, BET and TEM in the abstract; (b) Introduction: ZSM-5, SBA-15, UIO-66, NPs; (c) Materials and Methods: H2BDC, Pd@U
Reviewer 4 Report
This paper deals with the development of Pd/UIO-66 catalysts for the oxidation of toluene. There are numerous characterizations, the paper is well organized. The subject is not highly novel but the work is well-executed. It fits the scope of Materials. I recommended this publication after some corrections:
There are typos and the English needs to be polished, I suggest to read it by a native speaker. For example, a space is needed between the reference and the text (as [12] and not as[12]), L79 it is written uL, I suppose it is µL or mL, L100 and L230-231 -1 needs to be in exponent. In the introduction, it will be nice to have 2-3 sentences explaining the structure-composition of UIO-66 MOF. In the introduction (L64), you said “…is characterized by the Brunauer-Emmett-Teller (BET)…”, the materials are characterized by nitrogen adsorption-desorption isotherms and BET model is applied on the data, so you should modify your sentence. L79, what is the solvent for the Pd solution (hexane also?). L111, for me, it is physico-chemical properties not physical properties. L114-115, no hysteresis is observed on figure 1. Either you removed this sentence either you zoom on the mentioned zone in figure 1. For Figure 3, it would be nice if you can measure the Pd particle size from the pictures to give an estimation of the different sizes for the samples. L193, you mentioned the LH model, it is better to define the acronym (Langmuir–Hinshelwood) then use LH. Concerning the catalytic experiments, there is no information about the duration to obtain each point of Figure 5A. For the stability study, what is the temperature of the long time experiment of 20h? Your paper will have a better impact if you compare you catalytic results with the literature to show the interest of your study.Author Response
Please see the attachment

Round 2
Reviewer 1 Report
The paper could be accepted in present form
Reviewer 2 Report
Publish
Reviewer 4 Report
Dear authors,
Thank you for your comments, the paper can be published.
Best regards